# Contextual Reserve Price Optimization in Auctions via Mixed-Integer Programming

**Joey Huchette**
Rice University
joehuchette@rice.edu

**Haihao Lu**
University of Chicago
haihao.lu@chicagobooth.edu

**Hossein Esfandiari**
Google Research
esfandiari@google.com

**Vahab Mirrokni**
Google Research
mirrokni@google.com

## Abstract

We study the problem of learning a linear model to set the reserve price in an auction, given contextual information, in order to maximize expected revenue from the seller side. First, we show that it is not possible to solve this problem in polynomial time unless the *Exponential Time Hypothesis* fails. Second, we present a strong mixed-integer programming (MIP) formulation for this problem, which is capable of exactly modeling the nonconvex and discontinuous expected reward function. Moreover, we show that this MIP formulation is ideal (i.e. the strongest possible formulation) for the revenue function of a single impression. Since it can be computationally expensive to exactly solve the MIP formulation in practice, we also study the performance of its linear programming (LP) relaxation. Though it may work well in practice, we show that, unfortunately, in the worst case the optimal objective of the LP relaxation can be $\mathcal{O}(\text{number of samples})$ times larger than the optimal objective of the true problem. Finally, we present computational results, showcasing that the MIP formulation, along with its LP relaxation, are able to achieve superior in- and out-of-sample performance, as compared to state-of-the-art algorithms on both real and synthetic datasets. More broadly, we believe this work offers an indication of the strength of optimization methodologies like MIP to exactly model intrinsic discontinuities in machine learning problems.

## 1 Introduction

Digital advertising is a tremendously fast growing industry: the worldwide digital advertising expenditure was $283 billion in 2018, and it is estimated to further grow to $517 billion in 2023.[1]

Real time bidding (RTB) stands out as one of the most significant developments of the past decade in the space of advertisement allocation mechanisms, as it is widely utilized by major online advertising platforms including–but not limited to–Google, Facebook, and Amazon. In RTB for display ads, a user visiting a webpage instantaneously triggers an auction held by an Ad Exchange, wherein the winner of the auction earns the ad slot and pays the publisher a certain price.

A form of auction commonly used in practice by Ad Exchanges is a second-price auction with reserve price [22]. In such auctions, the publisher or Ad Exchange sets a reserve price before the auction is held, and the highest bidder wins the ad slot and pays the maximum of the second price and the

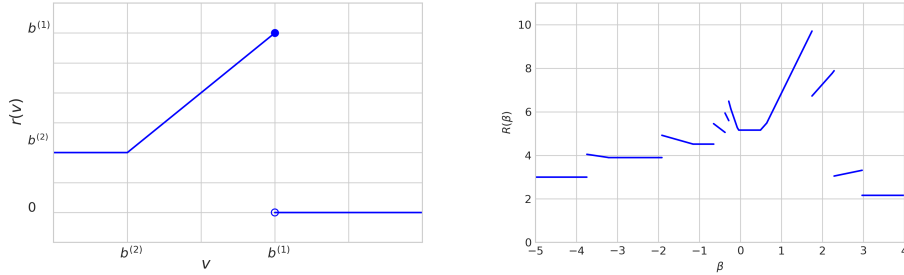

Figure 1: **(Left)** The revenue function $r(v; b^{(1)}, b^{(2)})$. **(Right)** Average revenue function $R(\boldsymbol{\beta})$ with $d = 1$ features and $n = 8$ samples.

reserve price. Reserve prices can increase revenue if they are set between the top two bids, but can also lead to a failed auction if set too high.

One central question for Ad Exchanges is *how to set the reserve price for each incoming impression in order to maximize the total revenue*. In general, the reserve price is set based on the contextual information of the ad campaign, including publisher data (e.g. ad site and ad size), user data (e.g. device type and various geographic information), and time (e.g. date and hour). In this paper, we aim to learn an offline linear model to set the reserve price in order to maximize total revenue on the seller side, using available contextual information. We model this via the optimization problem

$$\max_{\boldsymbol{\beta} \in X} \quad R(\boldsymbol{\beta}) := \frac{1}{n} \sum_{i=1}^{n} r(\boldsymbol{w}^i \cdot \boldsymbol{\beta}; b_i^{(1)}, b_i^{(2)}), \tag{1}$$

where $b_i^{(1)}$ and $b_i^{(2)}$ are, respectively, the (nonnegative) highest bidding price and the second highest bidding price of impression $i$, $\boldsymbol{w}^i \in \mathbb{R}^d$ is the contextual feature vector of impression $i$, and $X = [L, U]^d \subset \mathbb{R}^d$ is a bounded hypercube which serves as a feasible region for the model parameters $\boldsymbol{\beta}$. Note that, by artificially modifying the problem data, (1) can readily recover a first price auction (by setting $b^{(2)} = b^{(1)}$) or a pure price-setting problem (by setting $b^{(2)} = 0$). Additionally, $r$ is the discontinuous reward function

$$r(v; b^{(1)}, b^{(2)}) := \begin{cases} b^{(2)} & v \le b^{(2)} \\ v & b^{(2)} < v \le b^{(1)} \\ 0 & v > b^{(1)} \end{cases} . \tag{2}$$

Figure 1 plots the reward function $r(v; b^{(1)}, b^{(2)})$, which is a simple univariate (though discontinuous) function for given bidding prices $b^{(1)}$ and $b^{(2)}$. If the reserve price is set below $b^{(2)}$, the auction reverts to a second price auction. If the seller manages to set the reserve price in the "sweet spot" between $b^{(2)}$ and $b^{(1)}$, the seller can capture additional revenue from the auction. However, the seller must be wary not to set the reserve price too high, as in this case the sale does not occur and the reward drops to zero. Note that the reserve price is set after the contextual information is observed, but before the bidding prices are observed, making this price setting problem nontrivial.

Although the univariate function $r(\cdot; b^{(1)}, b^{(2)})$ is simple, the average revenue function $R$ can be extremely complicated, even for small problem instances. Figure 1 plots the average revenue $R(\beta)$ over 8 samples as a function of a single feature $\beta \in \mathbb{R}$, randomly drawn from a log-normal distribution as specified in Section 4. As we can see in Figure 1, the average revenue function $R$ has many local maximizers and is discontinuous, even in the small-sample, univariate setting. This complexity will only be exacerbated in the large-sample, multivariate case which is the focus of this paper.

As the reserve price must be set before the auction is held, any proposed model must allow extremely fast inference. However, the model can be trained off-line and then updated at regular intervals (e.g. daily or weekly), meaning that learning need not happen in real time. We focus on linear models in this submission, as their simplicity, interpretability, and fast inference lend themselves exceedingly well to the RTB setting. However, the proposed formulation and techniques can also be extended to more complicated machine learning models, such as kernel methods and (optimal) decision trees.

Moreover, many RTB platforms support an incredibly large number of auctions, meaning that an enormous amount of training data is available. This motivates learning algorithms which are highly scalable and, ideally, parallelizable.

## 1.1 Our Results

Our contribution in this work is threefold.

**Hardness (Section 2).** Our first main result is to build off the intuition gleaned from Figure 1 to show that (1) is, indeed, a hard problem. In particular, we show that there is no algorithm that solves (1) in polynomial time unless the Exponential Time Hypothesis fails. The Exponential Time Hypothesis is a very popular assumption is computational complexity concerning the 3-SAT problem [31], and it is the basis of many hardness results [1, 10, 11, 13, 18, 29, 34, 36, 42]. The Exponential Time Hypothesis states that 3-SAT can not be solved in subexponential time in the worst case. In order to show this result, we reduce our problem to the classic $k$-densest subgraph problem.

**New algorithms (Section 3).** Our second main result is an exact model for the problem using Mixed-Integer Programming (MIP). MIP is an optimization methodology capable of modeling complex, nonconvex feasible regions, and which is widely used in practice. In particular, MIP allows us to *exactly* model the underlying discontinuous reward function, without relying on convex or continuous proxies which may be poor approximations or require careful hyperparameter tuning.

One issue with MIP is that it is not scalable beyond medium-sized instances, and so is it cannot be brought to bear on problem instances with millions of past observations. In order to deal with the large-scale problems in daily auctions, we propose a Linear Programming (LP) relaxation of our MIP formulation. Modern LP solvers, such as Gurobi, are capable of solving very large LPs with millions of variables. The solution to the LP not only provides a valid upper bound to the optimal expected revenue, but can also yield feasible solutions to (1). While these solutions are often of relatively good quality, we show that in the worst case the LP relaxation can produce arbitrarily bad bounds on the true optimal reward.

**Computational validation (Section 4).** Finally, we present a thorough computational study on both synthetic and real data. We start with a low-dimensional artificial data set where we observe that existing methods, while exhibiting low generalization error, are substantially outperformed by our approaches. We also study a real data set comprised of eBay sports memorabilia auctions, where we observe a consistent improvement of our MIP-based methods over existing techniques. In both studies, we observe that our MIP formulation substantially outperforms the LP relaxation, its convex counterpart, suggesting the merit of using principled nonconvex approaches for this problem.

## 1.2 Related Work

**Reserve price optimization.** Reserve price optimization has been widely studied in both academia and industry due to its critical role in online advertisement. The main departure of our setting from much of the literature in this area is our explicit incorporation of the contextual information $\boldsymbol{w}$ into the optimization. Most previous theoretical works proceed under the assumption that the bidding prices come from a certain distribution without the consideration of contextual information. For example, [12] shows a regret minimization under the assumption that all bids are independently drawn from the same unknown distribution; [30] shows the constant reserve is optimal when the distribution is known and satisfies certain regularity assumptions; and [2] studies the case when the buyers are strategic and would like to maximize their long-term surplus.

In practice, however, an Ad Exchange logs contextual information of every auction and uses this data to determine future reserve price. For example, in a large field study at Yahoo! [39], contextual information was used to learn the bidding distribution of buyers, which was then use to set up the future reserve price. This is an indirect use of contextual information. In contrast, (1) builds a linear model for reserve price optimization by directly using the contextual information.

To the best of our knowledge, the only work which directly uses the contextual information to set up the reserve price is that of Mohri and Medina [38]. In order to handle the discontinuity in the revenue function $r$, [38] present a continuous piecewise linear surrogate function, and optimize over this surrogate function using difference-of-convex programming. There are several difficulties of the method proposed in [38]: (i) it is non-trivial to tune the hyper-parameter $\gamma$ in the surrogate function,

which controls the closeness of the two problems and the hardness to solve the surrogate problem; (ii) the global convergence of difference-of-convex programming is slow (requiring, e.g., a cutting plane or branch-and-bound method) and requires a careful implementation [26], and (iii) it can only find a local optimizer of the surrogate problem. In contrast, we directly solve the reserve price optimization problem (1) by mixed-integer programming.

**Mixed-integer programming for piecewise linear functions.** Mixed-integer programming has long been used to model piecewise linear functions in a number of application areas as disparate as operations [16, 17, 33], analytics [7, 8], engineering [23, 24], and robotics [19, 20, 32, 37]. In this literature, our approach is most related to a recent strain of approaches applying MIP to model high-dimensional piecewise linear functions arising as trained neural networks for various tasks such as verification and reinforcement learning [4, 3, 40, 41]. Moreover, there are sophisticated and mature implementations of algorithms for mixed-integer programming (i.e. *solvers*) that can reliably solve many instances of practical interest in reasonable time frames.

**Hardness.** We study the hardness of the reserve price optimization problem (1) and show that it is impossible to solve in polynomial time unless the *Exponential Time Hypothesis* [27] fails. The exponential time hypothesis is a very popular assumption in computational complexity and it is the basis for many hardness results such as approximating the best Nash equilibrium [11], $k$-densest subgraph [10, 27], SVP [1], network design [14], and many others [34, 42, 29, 13, 18, 36].

## 2 Hardness

In this section we show the hardness of the reserve price optimization problem (1). Specifically, we show that it is not possible to solve this problem in polynomial time unless the Exponential Time Hypothesis fails. We prove this by showing that a polynomial time optimal algorithm for this problem implies a polynomial time constant approximation algorithm for the $k$-*densest subgraph problem*.

**Definition 1** ($k$-densest subgraph problem)**.** *Take a graph $G = (V_G, E_G)$, where $V_G$ represents the vertex set and $E_G$ represents the edge set. The goal is to find a subgraph $H = (V_H, E_H) \subseteq G$ with $|V_E| = K$ that maximizes $\frac{|E_H|}{|V_H|}$ .*

There is no $|V_G|^{-1/\operatorname{poly}(\log\log|V_G|)}$- approximation polynomial time algorithm for the $k$-densest subgraph problem unless the exponential time hypothesis fails [36]. Therefore, we produce a reduction that gives the following result.

**Theorem 1.** *There is no polynomial time algorithm for the reserve price optimization problem* (1)*, unless the Exponential Time Hypothesis fails.*

The proofs for Theorem 1, and all other technical results, are deferred to the Appendix.

## 3 New formulations

In this section, we develop a mixed-integer programming (MIP) formulation for solving (1), study its important computational properties, and discuss how to use it in practice.

MIP is an common optimization methodology capable of modeling complex, nonconvex constraints. MIP formulations comprise a set of linear constraints in the decision variables, along with integrality constraints on some (or all) of the variables.

In order to model (1) with MIP, we first start with the graph of the revenue function $r(\cdot; b^{(1)}, b^{(2)})$, which is defined as $\operatorname{gr}(r(\cdot; b^{(1)}, b^{(2)}); D) := \left\{ (v, y) \mid v \in D, \ y = r(v; b^{(1)}, b^{(2)}) \right\}$. This set is not closed, due to the discontinuity of $r$ at input $b^{(1)}$. Nonetheless, (1) can be reformulated using closures:

$$\max_{\boldsymbol{\beta}, \boldsymbol{v}, \boldsymbol{y}} \quad \frac{1}{n} \sum_{i=1}^{n} y_i \tag{3a}$$

$$\text{s.t.} \quad v_i = \boldsymbol{w}^i \cdot \boldsymbol{\beta} \qquad\qquad \forall i \in [\![n]\!] \tag{3b}$$

$$(v_i, y_i) \in \operatorname{cl}(\operatorname{gr}(r(\cdot; b_i^{(1)}, b_i^{(2)}); [l_i, u_i])) \quad \forall i \in [\![n]\!] \tag{3c}$$

$$\boldsymbol{\beta} \in X, \tag{3d}$$

where the bounds on the $v$ variables are computed as $l_i := \min_{\boldsymbol{\beta} \in X} \boldsymbol{w}^i \cdot \boldsymbol{\beta}$ and $u_i := \max_{\boldsymbol{\beta} \in X} \boldsymbol{w}^i \cdot \boldsymbol{\beta}$. It is straightforward to add a learned constant offset term $\beta_0$ to the model by changing (3b) to $v_i = \boldsymbol{w}^i \cdot \boldsymbol{\beta} + \beta_0$, though we omit it for the remainder of the section for notational simplicity.

**Proposition 1.** *If a point $(\boldsymbol{\beta}, \boldsymbol{v}, \boldsymbol{y})$ is an optimal solution for (3), then $\boldsymbol{\beta}$ is an optimal solution for (1). Conversely, if $\boldsymbol{\beta}$ is an optimal solution for (1), then there exists some $\boldsymbol{v}$ and $\boldsymbol{y}$ such that $(\boldsymbol{\beta}, \boldsymbol{v}, \boldsymbol{y})$ is an optimal solution for (3).*

We can now construct a mixed-integer programming formulation for (3c).

**Proposition 2.** *A valid MIP formulation for the constraint*

$$(v, y) \in \mathrm{cl}(\boldsymbol{gr}(r(\cdot; b^{(1)}, b^{(2)}); [l, u])) \tag{4}$$

*is:*

$$y \le b^{(2)} z_1 + b^{(1)} z_2, \qquad y \ge b^{(2)}(z_1 + z_2) \tag{5a}$$

$$y \le v + (b^{(2)} - l) z_1 - b^{(1)} z_3, \quad y \ge v - u z_3 \tag{5b}$$

$$l \le v \le u \tag{5c}$$

$$z_1 + z_2 + z_3 = 1, \quad \boldsymbol{z} \in [0, 1]^3 \tag{5d}$$

$$\boldsymbol{z} \in \mathbb{Z}^3. \tag{5e}$$

Piecing it all together, we can present a MIP formulation for the original problem (1).

**Corollary 1.** *Take $F(b^{(1)}, b^{(2)}, l, u)$ as the set of all points feasible for (5), given data $b^{(1)}$, $b^{(2)}$, $l$, and $u$. Modify (3) by, for each $i \in [\![n]\!]$, replacing the constraint (3c) with the constraint $(v_i, y_i) \in F(b_i^{(1)}, b_i^{(2)}, l_i, u_i)$; call this modification $\mathtt{MIP}$. Then (1) is equivalent to $\mathtt{MIP}$ in the sense that: (i) if $(\boldsymbol{\beta}, \boldsymbol{v}, \boldsymbol{y})$ is an optimal solution to $\mathtt{MIP}$, then $\boldsymbol{\beta}$ is an optimal solution to (1), and (ii) if $\boldsymbol{\beta}$ is an optimal solution to (1), then there exists some $\boldsymbol{v}$ and $\boldsymbol{y}$ such that $(\boldsymbol{\beta}, \boldsymbol{v}, \boldsymbol{y})$ is an optimal solution to $\mathtt{MIP}$.*

### 3.1 The tightness of formulation (5)

One measure of the quality of a MIP formulation is how tightly its LP relaxation approximates the set it formulates. MIP formulations with tight relaxations are likely to solve much more quickly than those with looser relaxations. The tightest possible MIP formulation is *ideal*, wherein all extreme points of the LP relaxation are integral [43]. The next proposition shows that (5) is ideal for (4).

**Proposition 3.** *The MIP formulation (5) is ideal for (4), in the sense that the linear programming relaxation (5a-5d) is a description of the convex hull of all $(v, y, \boldsymbol{z})$ feasible for (5).*

### 3.2 The feasible region

While the statement of the problem (1) constrains the model parameters $\boldsymbol{\beta}$ to lie within a bounded hypercube, it may be difficult to infer the correct size of the domain a priori. To illustrate, we present a low-dimensional family of instances where the problem data is bounded in magnitude, but nevertheless the magnitude of the optimal model parameters goes to infinity.

**Proposition 4.** *Fix $n = 2$ samples and $d = 2$ features, and consider $X = \mathbb{R}^2$, i.e. the unbounded variant of (1). There exists a sequence of instances where the problem data is bounded in magnitude by one, and yet the magnitude of the unique optimal solution to (1) grows arbitrarily large.*

In other words, we cannot bound the magnitude of the components of an optimal solution solely as a function of $n$, $d$, and the magnitude of the data. However, due to existential representability results [28], applying MIP formulation techniques to model (1) will require a bounded domain $X$ on the model parameters. To circumvent this, we model the magnitude of the bounding box as a hyperparameter, and tune it using a validation data set. This is the same approach taken in the difference-of-convex algorithm due to Mohri and Medina [38].

### 3.3 The linear programming relaxation

Our MIP formulation (5) comprises two types of constraints: linear constraints (5a-5d), and integrality constraints (5e). The linear programming relaxation comprises only the linear constraints, and

provides a valid dual upper bound on the optimal reward of a linear programming formulation. Furthermore, for this particular problem, each feasible solution for the linear programming relaxation corresponds to a feasible solution for the original problem (1).

**Proposition 5.** *Take $W(b^{(1)}, b^{(2)}, l, u)$ as the set of all points feasible for the LP relaxation (5a-5d) of (4), given data $b^{(1)}$, $b^{(2)}$, $l$, and $u$. Modify (3) by, for each $i \in [\![n]\!]$, replacing the constraint (3c) with the constraint $(v_i, y_i) \in W(b_i^{(1)}, b_i^{(2)}, l_i, u_i)$; call this modification* LP. *Then* LP *is an LP relaxation of (1) in the sense that the optimal reward for* LP *upper bounds the reward of any feasible solution for (1). Moreover, for any feasible solution $(\boldsymbol{\beta}, \boldsymbol{v}, \boldsymbol{y})$ to* LP, $\boldsymbol{\beta}$ *is a feasible solution to (1).*

Therefore, a third approach to solve (1) is simply to solve the linear programming relaxation. Linear programming problems can be solved in polynomial time, and there exist algorithms that can very efficiently solve large scale problem instances. Therefore, the approach of Proposition 5 can be applied to very large scale instances of the problem (1).

In practice, the Ad Exchange usually processes millions of impressions per minute, and updates the model parameters frequently (say, every 10 minutes) by learning from the data in the past time period. In this large-scale setting, MIP-based algorithms or the approach of Mohri and Medina [38] are not viable. Fortunately, the LP relaxation method remains viable, as modern Linear Programming solvers, such as Gurobi or CPLEX, can often solve huge LP with millions of variables within minutes.

A corollary of Proposition 3 is that, if $n = 1$, the LP relaxation LP is exact, and so exactly represents the convex hull of feasible points for MIP. Unfortunately, the composition of ideal formulations will, in general, fail to be ideal. In fact, the the optimal reward from LP can be arbitrarily bad as $n$ grows.

**Proposition 6.** *There is a family of instances of (1), parameterized by the sample size $n$, where the optimal reward of (1) decreases as $1/n$, but the optimal reward for the LP relaxation* LP *is at least* 1.

## 4 Computational study

We now perform a computational study on our proposed methods, using both synthetic and real data.

### 4.1 Implementation details

**Methods.** Throughout, we compare seven methods:

1. CP: $\max_v \frac{1}{n} \sum_{i=1}^n r(v; b_i^{(1)}, b_i^{(2)})$ – The optimal constant reserve price policy (i.e, set the reserve price to a constant for all samples without using contextual information). It is used as a benchmark to measure the improvement to be gained from using contextual information.

2. LP: The linear programming relaxation presented in Proposition 5.

3. MIP: The MIP formulation of Corollary 1 terminated after a time limit (to be specified subsequently).

4. MIP-R: The MIP formulation of Corollary 1 terminated at the root node[2].

5. DC: The difference-of-convex algorithm of Mohri and Medina [38].

6. GA: Gradient ascent, with a strong Wolfe line search.

7. UB: $\frac{1}{n} \sum_{i=1}^n b_i^{(1)}$ – This is a perfect information upper bound equal to the average first bid price. This is the largest reward that can possibly be garnered from the auction. Note that this may be quite a loose upper bound, as in general there will not exist a linear model capable of setting such reserve prices given the contextual information.

**Hyperparameter tuning.** The LP, MIP-R, and MIP algorithms require that the parameter domain $X$ is explicitly specified. We utilize cross validation to tune the bounds on each parameter as $[-T, +T]$ for $T \in \{2^{-1}, \ldots, 2^9\}$. Additionally, DC requires two hyperparameters: one for a penalty associated with the bound constraints, and the second for the "slope" of its continuous approximation of the discontinuous reward function $r$. We do cross-validation as suggested in Mohri and Medina [38].

**Evaluation.** For each experiment, we report the *average reward* (i.e. $R(\boldsymbol{\beta})$) of the final model from each algorithm on both the training and test data sets. We also use the "gap closed" metric to measure the improvement of `MIP` over `DC`, the best existing algorithm from the literature. This is computed as as $\frac{\texttt{MIP}-\texttt{DC}}{\texttt{UB}-\texttt{DC}}$, where in an abuse of notation we use the algorithm names to denote their respective rewards.

**Implementation.** We implement our experiment in Julia [9]. We use JuMP [21, 35] and Gurobi v8.1.1 [25] to model and solve, respectively, the optimization problems underlying the `MIP`, `MIP-R`, `LP`, and `DC` methods. Our implementation is publicly available at: `https://github.com/joehuchette/reserve-price-optimization`.

## 4.2 Synthetic data

**Data generation.** Here we describe how we generate our synthetic data $(\boldsymbol{w}^i, b_i^{(1)}, b_i^{(2)})_{i=1}^n$. First, the feature vectors $\boldsymbol{w}^i$ are generated i.i.d. from a Gaussian distribution with identity covariance matrix, i.e., $\boldsymbol{w}^i \stackrel{iid}{\sim} d^{-1/2} \mathcal{N}(0, I^d)$, normalized so that $\mathbb{E}\|\boldsymbol{w}^i\|_2^2 = 1$. In order to generate the bidding prices $b_i^{(1)}$ and $b_i^{(2)}$, we assume there are two buyers, and they have underlining generative parameters $\boldsymbol{c}_1$ and $\boldsymbol{c}_2$, such that their bids come from log-normal distributions as $b_1^i \stackrel{iid}{\sim} \mathcal{LN}(\boldsymbol{c}_1 \cdot \boldsymbol{w}^i, \sigma|\boldsymbol{c}_1 \cdot \boldsymbol{w}^i|)$ and $b_2^i \stackrel{iid}{\sim} \mathcal{LN}(\boldsymbol{c}_2 \cdot \boldsymbol{w}^i, \sigma|\boldsymbol{c}_2 \cdot \boldsymbol{w}^i|)$, where $\sigma$ controls the signal-to-noise ratio of the log-normal distribution. We then set $b_i^{(1)} = (1 + \alpha) \max\{b_1^i, b_2^i\}$ and $b_i^{(2)} = (1 - \alpha) \min\{b_1^i, b_2^i\}$, where $\alpha$ is a dilation factor to enlarge the difference between $b_i^{(1)}$ and $b_i^{(2)}$.[3] Moreover, the underlying parameters $\boldsymbol{c}_1$ and $\boldsymbol{c}_2$ of the two buyers should be correlated, since the bidding prices for high-valued slots should be high for all buyers. In order to model this, we set $\boldsymbol{c}_1 = \boldsymbol{h}_1$ and $\boldsymbol{c}_2 = \rho \boldsymbol{h}_1 + \sqrt{1 - \rho^2}\boldsymbol{h}_2$, where $\boldsymbol{h}_1, \boldsymbol{h}_2 \stackrel{iid}{\sim} d^{-1/2}\mathcal{N}(0, I^d)$ and $\rho$ controls the correlation between $\boldsymbol{c}_1$ and $\boldsymbol{c}_2$. We normalize the bid prices so that the mean first price is 1.

Overall, we have three parameters in the data generation process: $\sigma$ controls the signal-to-noise level of the model, $\rho$ controls the similarity between two buyers, and $\alpha$ controls the degree of flexibility the seller has when setting a reserve price.

**Experimentation.** We fix $d = 50$ features, $n = 1000$ training samples, along with test and validation data sets each with 5000 samples. We first set a "baseline" configuration for our generative model with $\sigma = 0.1$, $\rho = 0.9$, and $\alpha = 0.1$. To explore the robustness of our model to changes in the data generation scheme, we then study three variants of this baseline with "high noise" ($\sigma = 0.5$), "low correlation" ($\rho = 0.5$), and "low margin" ($\alpha = 0.02$). For each of these four parameter settings, we give aggregate results over three trials in Table 1. We set a time limit of 3 minutes for each algorithm.

In all four experiments, `MIP` offers an improvement over `DC`, typically considerably so. On the baseline configuration, `MIP` closes an average of 83.5% of the gap left by `DC` on the training set, and 66.2% of the gap left remaining on the test set. Unsurprisingly, the high noise configuration leads to degradation of performance with respect to the perfect information upper bound, but `MIP` is still able to close 65.5% and 47.4% of the gap on the training and test data sets, respectively. The low correlation configuration sees `MIP` closing 79.0% and 61.5% of the remaining gap on training and test data sets, respectively, while on the low margin configuration `MIP` closes 60.5% of training gap and 16.3% of test gap.

While `MIP-R` does not quite attain the same level of performance as `MIP`, it is quite close and still generally outperforms `DC` both in- and out-of-sample. The `LP` method also outperforms `DC` on three of four experiments, albeit by a smaller margin. We observe that `DC` produces models that lead to sales on nearly every impression. In contrast, the `LP` algorithm sets reserve prices too aggressively, leading to a model that set reserve prices that lead to failed auctions on roughly 10-20% of impressions. Additionally, we observe that the `MIP` and `MIP-R` methods both produce models that yield sales on nearly all impressions. This indicates that they are not exploiting a small number of impressions that garner a high reward, but instead are intelligently setting a reserve price policy that captures excess reward across the population, without too aggressively setting the prices so that many impressions fail to sell.

| method | train | test |
|--------|-------|------|
| CP | 0.790 ±0.007 | 0.788 ±0.002 |
| LP | 0.854 ±0.005 | 0.808 ±0.011 |
| MIP | 0.962 ±0.006 | 0.924 ±0.003 |
| MIP-R | 0.962 ±0.006 | 0.923 ±0.002 |
| DC | 0.776 ±0.006 | 0.776 ±0.002 |
| GA | 0.516 ±0.516 | 0.518 ±0.518 |
| UB | 0.999 ±0.006 | 1.000 ±0.001 |

(a) Baseline.

| method | train | test |
|--------|-------|------|
| CP | 0.783 ±0.015 | 0.777 ±0.020 |
| LP | 0.810 ±0.008 | 0.774 ±0.025 |
| MIP | 0.907 ±0.013 | 0.858 ±0.017 |
| MIP-R | 0.832 ±0.004 | 0.792 ±0.026 |
| DC | 0.752 ±0.004 | 0.750 ±0.007 |
| GA | 0.395 ±0.418 | 0.386 ±0.419 |
| UB | 0.998 ±0.004 | 1.000 ±0.001 |

(b) High noise.

| method | train | test |
|--------|-------|------|
| CP | 0.785 ±0.026 | 0.781 ±0.026 |
| LP | 0.798 ±0.037 | 0.766 ±0.042 |
| MIP | 0.942 ±0.010 | 0.889 ±0.014 |
| MIP-R | 0.943 ±0.087 | 0.889 ±0.013 |
| DC | 0.733 ±0.011 | 0.730 ±0.013 |
| GA | 0.488 ±0.479 | 0.484 ±0.480 |
| UB | 1.001 ±0.003 | 0.999 ±0.001 |

(c) Low correlation.

| method | train | test |
|--------|-------|------|
| CP | 0.911 ±0.003 | 0.910 ±0.003 |
| LP | 0.917 ±0.003 | 0.890 ±0.004 |
| MIP | 0.964 ±0.002 | 0.921 ±0.007 |
| MIP-R | 0.911 ±0.003 | 0.910 ±0.003 |
| DC | 0.911 ±0.003 | 0.910 ±0.003 |
| GA | 0.708 ±0.220 | 0.706 ±0.222 |
| UB | 1.001 ±0.001 | 0.999 ±0.001 |

(d) Low margin.

Table 1: Synthetic data results.

Finally, GA does very poorly in all settings with quite high variance of the performance. This makes intuitive sense, as gradient information is less informative when the objective is discontinuous.

### 4.3 eBay auctions for sports memorabilia

We now turn our attention to a real data set. We use a published medium-size eBay data set for reproducibility, which comprises 70,000 sports memorabilia auctions, to illustrate the performance of our algorithms. The data set is provided by Jay Grossman and was subsequently studied in the context of reserve price optimization [38].[4] There are 78 features in the data, with both seller information (e.g. rating and location) and item information. We preprocess the data by normalizing the bidding prices with the mean of their first prices.

Table 2 depicts the average and the 95% confidence interval of the cumulative reward on both training and test data set over 10 random runs. In both, we use 2000 randomly selected samples from the data set for testing and for validation. In Table 2a, we train using 2000 randomly selected samples and a time limit of 5 minutes, while in Table 2b we utilize 5000 training samples and a time limit of 15 minutes.

In Table 2, MIP outperforms all other methods, producing the best performing models as measured on both the training and test data sets. The DC algorithm is the next best performer, producing higher quality models than both LP and MIP-R. Indeed, MIP closes 7.39% of the gap left by DC on the training data set, with respect to the UB upper bound. However, due to a lack of generalization, this number shrinks considerably to 1.66% on the test data set. There is no doubt that DC has a smaller generalization gap, although one plausible explanation for this could be the additional hyperparameters tuned over in the DC method. Moreover, we emphasize that these gaps are computed based on a conservative upper bound (i.e., UB) which, as observed in Section 4.2, may be quite loose.

In order to understand the behavior of the algorithms on larger data sets, we increase the training data sample size to 5000 and repeat the eBay experiments. The results are depicted in Table 2b. While the rankings of the algorithms remains the same, MIP is able to extract more information from the larger data set. The training reward grows, and the models produced also generalize much more successfully to the testing data set. In contrast, the DC algorithm appears unable to exploit the extra available data, with training and test accuracy that remain nearly identical with the previous experiment. Indeed, MIP is able to close 9.11% of the remaining gap on the training data set, and 7.01% on the testing data set.

| method | train | test | method | train | test |
|--------|-------|------|--------|-------|------|
| CP | 0.563 $\pm0.007$ | 0.568 $\pm0.010$ | CP | 0.564 $\pm0.004$ | 0.567 $\pm0.023$ |
| LP | 0.668 $\pm0.006$ | 0.654 $\pm0.020$ | LP | 0.665 $\pm0.006$ | 0.650 $\pm0.016$ |
| MIP | 0.726 $\pm0.009$ | 0.714 $\pm0.015$ | MIP | 0.731 $\pm0.009$ | 0.725 $\pm0.013$ |
| MIP-R | 0.657 $\pm0.022$ | 0.652 $\pm0.027$ | MIP-R | 0.596 $\pm0.038$ | 0.596 $\pm0.041$ |
| DC | 0.704 $\pm0.007$ | 0.709 $\pm0.016$ | DC | 0.704 $\pm0.007$ | 0.704 $\pm0.015$ |
| GA | 0.396 $\pm0.165$ | 0.398 $\pm0.165$ | GA | 0.363 $\pm0.164$ | 0.363 $\pm0.261$ |
| UB | 0.992 $\pm0.006$ | 1.014 $\pm0.018$ | UB | 1.002 $\pm0.006$ | 0.999 $\pm0.023$ |

(a) 2000 training samples.      (b) 5000 training samples.

Table 2: Ebay data set experiments.

Comparing Table 2a and Table 2b, we can clearly see that the difference in reward produced by MIP between the training and test data sets decreases as number of samples increases. This is intuitively consistent with what could be expected from a learning theory analysis, and we expect that this gap will likely keep shrinking in the "big data" regime as we further enlarge the training sample size.

# 5 Conclusion and Future Directions

In this paper, we study the linear model for reserve price optimization in a second-price auction. We first show that this is indeed a hard problem – unless the Exponential Time Hypothesis fails, there is no polynomial time optimal algorithm. Then we propose a mixed-integer programming formulation and a LP relaxation for solving the problem. Linear models are the simplest learning model for this problem with fast inference and straightforward interpretability. How to extend our approaches developed herein to other learning methods, such as kernel methods and optimal decision trees, are solid future research directions.

## Broader Impact

This work presents new methods, and as such does not have direct societal impact. However, if the context provided allows the model to reason about protected classes or sensitive information, either directly or indirectly, the model–and, therefore, the application of this work–has the potential for adverse effects.

## Funding Disclosure

No third-party funding was received for this work.

## Footnotes

[1]"Digital advertising spending worldwide 2018-2023", retrieved May 25 2020 from https://www.statista.com/statistics/237974/online-advertising-spending-worldwide/

[2]This means the solver will terminate just before begining its enumerative tree search procedure. It will solve the LP relaxation, but crucially will also run a bevy of heuristics to improve primal solutions and dual bounds that require the knowledge that the underlying model is a MIP.

[3] We note that this dilation is similar to the scaling of linear functions used in the generative model of [38].

[4]"Ebay Data Set", accessed May 25 2020 from https://cims.nyu.edu/~munoz/data/. We refer the reader to [38] for a more detailed description of the data set.

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
