[Supplementary Material]

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

# A  Deferred Proofs

## A.1  Proof of Theorem 1

*Proof.* Let $G = (V_G, E_G)$ be an arbitrary input graph to the $k$-densest subgraph problem, where $V_G$ is the vertex set of the graph and $E_G$ is the edge set of the graph. We construct an input to the reserve price optimization problem (1) based on $G$, so that if it were possible to solve the reserve price optimization problem for this input in polynomial time, this would imply that it is possible to find an $1/8$-approximate solution to the $k$-densest subgraph problem on $G$ in polynomial time. However, it is known that it is impossible to give a polynomial time $1/8$ approximation algorithm for the densest subgraph problem unless the exponential time hypothesis fails [36]. This implies that it is impossible to solve the reserve price optimization problem (1) unless the exponential time hypothesis fails.

Next, we explain how to construct an input to the reserve price optimization problem (1) based on $G$. In the optimization problem we set $X = [0,1]^d$. We have two types of impressions as explained below.

- We have $|V_G|^2$ impressions $(\boldsymbol{w}_1, k, 0)$, where $\boldsymbol{w}_1 = \langle 1, 1, \dots, 1 \rangle$.

- For each edge $e = (u, v) \in E_G$, we have one impression $(\boldsymbol{w}_e, 2, 1.5)$, where $\boldsymbol{w}_e$ is a feature vector in which the components corresponding to $v$ and $u$ are 1, and all other components are 0.

First, we lower bound the optimal solution of the optimization problem (1) for this input. Consider a densest subgraph $H = (V_H, E_H)$ of $G$, where $V_H$ is the vertex set of $H$ and $E_H$ is the edge set of $H$. We define $\boldsymbol{\beta}_H$ to be a feature vector in which the features corresponding to the vertices of $V_H$ are 1, and all other features are 0. Next we bound $R(\boldsymbol{\beta}_H)$. We use this as a lower bound the optimum solution of the optimization problem (1).

Note that $\boldsymbol{w}_1 \cdot \boldsymbol{\beta}_H = k$, and hence the contribution of each of the first type of impressions to $R(\boldsymbol{\beta}_H)$ is $\frac{k}{n}$. Also, for each edge $e \in E_H$ we have $\boldsymbol{w}_e \cdot \boldsymbol{\beta}_H = 2$ and hence the contribution of each of the second type of impressions corresponding to an edge in $E_H$ to $R(\boldsymbol{\beta}_H)$ is $\frac{2}{n}$. Therefore, we have

$$R(\boldsymbol{\beta}_H) = \frac{1}{n}\Big( k|V_G|^2 + 1.5|E_G| + 0.5|E_H| \Big). \tag{6}$$

Next, we upper bound the optimal solution of the optimization problem (1) for our input. Let $\boldsymbol{\beta} = \langle \beta_1, \dots, \beta_{V_G} \rangle$ be the vector that maximizes $R(\boldsymbol{\beta})$. Note that if $\sum_v \beta_v > k$, the contribution of the first type of impressions is 0. This means that $R(\boldsymbol{\beta}) \le 2|E_G| < R(\boldsymbol{\beta}_H)$, which is a contradiction. Therefore, without loss of generality we can assume that $\sum_i \beta_i \le k$.

Let $V^{\boldsymbol{\beta}}$ be the set of vertices in $V_G$ with $\beta_v \ge 0.5$. Let $G^{\boldsymbol{\beta}} = (V^{\boldsymbol{\beta}}, E^{\boldsymbol{\beta}})$ be the subgraph of $G$ induced by $V^{\boldsymbol{\beta}}$. Note that if for a vertex $v$ we have $\beta_v < 0.5$, then for each edge $e = (v, u)$ neighboring $v$, we have $\boldsymbol{w}_e \cdot \boldsymbol{\beta}_H \le 1 + 0.5 = 1.5$. Therefore, we have

$$R(\boldsymbol{\beta}) \le k|V_G|^2 + 1.5|E_G| + 0.5|E^{\boldsymbol{\beta}}|. \tag{7}$$

Now, we put inequalities (6) and (7) together to complete the proof. By the optimality of $\boldsymbol{\beta}$ we have $R(\boldsymbol{\beta}_H) \le R(\boldsymbol{\beta})$. This together with inequalities (6) and (7) implies that $|E_H| \le |E^{\boldsymbol{\beta}}|$. Moreover, recall that for every vertex $v$ in $V^{\boldsymbol{\beta}}$ we have $\beta_v \ge 0.5$. Also, we have $\sum_i \beta_i \le k$. Hence, we have $|V^{\boldsymbol{\beta}}| \le 2k$. Given a graph with $2k$ vertices, one can easily cover the edges with 8 subgraphs of size $k$. By the pigeon hole principle one of these subgraphs contains $\frac{E^{\boldsymbol{\beta}}}{8} \ge \frac{E_H}{7}$ edges, and hence it is a $\frac{1}{8}$-approximate solution to the densest subgraph. $\qquad\square$

## A.2  Proof of Proposition 1

*Proof.* First, we show that each optimal solution for (1) has a corresponding feasible point for (3) with equal objective value. Take some $\boldsymbol{\beta}^*$ optimal for (1). Setting $v_i^* = \boldsymbol{w}^i \cdot \boldsymbol{\beta}^*$ for each $i$, the feasibility of $\boldsymbol{\beta}^*$ (i.e. $\boldsymbol{\beta}^* \in X$) implies that $l_i \le v_i^* \le u_i$ from the definition of $l_i$ and $u_i$. Now take

$y_i^* = r(v_i^*)$ for each $i$; clearly (3c) is satisfied. Therefore, $(\boldsymbol{\beta}^*, \boldsymbol{v}^*, \boldsymbol{y}^*)$ is feasible for (3) and has objective value $\frac{1}{n}\sum_{i=1}^n r(v_i; b_i^{(1)}, b_i^{(2)})$.

Next, we show that each optimal solution $(\boldsymbol{\beta}^*, \boldsymbol{v}^*, \boldsymbol{y}^*)$ for (3) corresponds to a feasible point $\boldsymbol{\beta}^*$ for (1) with the same objective value. Clearly $\boldsymbol{\beta}^*$ is feasible for (1). Additionally, (3c) means that for each $i$, if $v_i^* \neq b_i^{(1)}$ then $y_i^* = r(v_i^*)$, whereas if $v_i^* = b_i^{(1)}$ then $y_i^* \in \{r(v_i^*) \equiv b_i^{(1)}, 0\}$. As $b_i^{(1)} \geq 0$, the optimality of $(\boldsymbol{\beta}^*, \boldsymbol{v}^*, \boldsymbol{y}^*)$ implies that we must have $y_i^* = r(v_i^*)$. Therefore, the objective value of $(\boldsymbol{\beta}^*, \boldsymbol{v}^*, \boldsymbol{y}^*)$ is $\frac{1}{n}\sum_{i=1}^n r(v_i^*; b_i^{(1)}, b_i^{(2)})$, giving the result. □

## A.3 Proof of Proposition 2

*Proof.* Suppose $(\hat{\boldsymbol{v}}, \hat{\boldsymbol{y}}, \hat{\boldsymbol{z}})$ is feasible for (5). It follows from (5d–5e) that exactly one component of $\hat{z}$ is equal to one, with the other two components equal to zero. We now consider each of these three cases.

If $\hat{z}_1 = 1$, then the constraints (5a–5c) reduce to $y = b^{(2)}$, $v \leq y \leq v + b^{(2)} - l$, and $l \leq v \leq u$. Plugging in the equation for $y$ into the second pair of inequalities yields $l \leq v \leq b^{(2)}$, which are the constraints defining $S_1$.

If $\hat{z}_2 = 1$, the constraints (5a–5c) reduce to $b^{(2)} \leq y \leq b^{(1)}$, $y = v$, and $l \leq v \leq u$, which are equivalent to the constraints defining $S_2$.

Finally, if $\hat{z}_3 = 1$, the constraints (5a–5c) reduce to $y = 0$, $v - u \leq y \leq v - b^{(1)}$, and $l \leq v \leq u$. Plugging the equation into the pair of inequalities yields $b^{(2)} \leq v \leq u$, i.e. the constraints are equivalent to those defining $S_3$. □

## A.4 Proof of Proposition 3

**Lemma 1.** *The closure of $gr(r(\cdot; b^{(1)}, b^{(2)}); D)$ is $S_1 \cup S_2 \cup S_3$, where*

$$S_1 = \left\{ (v, y) \in D \times \mathbb{R} \;\middle|\; \begin{array}{l} y = b^{(2)} \\ v \leq b^{(2)} \end{array} \right\} \tag{8a}$$

$$S_2 = \left\{ (v, y) \in D \times \mathbb{R} \;\middle|\; \begin{array}{l} y = v \\ b^{(2)} \leq v \leq b^{(1)} \end{array} \right\} \tag{8b}$$

$$S_3 = \left\{ (v, y) \in D \times \mathbb{R} \;\middle|\; \begin{array}{l} y = 0 \\ v \geq b^{(1)} \end{array} \right\}. \tag{8c}$$

*Proof (Proposition 3).* Take $D$ as the set of all $(v, y, \boldsymbol{z})$ feasible for (5). Using Lemma 1, we can infer that $D = \bigcup_{i=1}^3 (S_i \times \{\mathbf{e}^i\})$, where $\mathbf{e}^i \in \{0, 1\}^3$ is the $i$-th unit vector of all zeros except a 1 in the $i$-th coordinate. Therefore, it can be expressed as a finite union of bounded polyhedron. Applying techniques due to Balas [5, 6], we can write a lifted representation for the convex hull of $D$, i.e. one

with auxiliary $v^i$ and $y^i$ variables:

$$v = \sum_{i=1}^{3} v^i \tag{9a}$$

$$y = \sum_{i=1}^{3} y^i \tag{9b}$$

$$y^1 = b^{(2)} z_1 \tag{9c}$$

$$l z_1 \leq v^1 \leq b^{(2)} z_1 \tag{9d}$$

$$y^2 = v^2 \tag{9e}$$

$$b^{(2)} z_2 \leq v^2 \leq b^{(1)} z_2 \tag{9f}$$

$$y^3 = 0 \tag{9g}$$

$$b^{(1)} z_3 \leq v^3 \leq u z_3 \tag{9h}$$

$$1 = z_1 + z_2 + z_3 \tag{9i}$$

$$\boldsymbol{z} \in [0, 1]^3. \tag{9j}$$

Moreover, if $R$ is the set of all points feasible for (9), it is known that $\mathrm{Proj}_{v,y,\boldsymbol{z}}(R) = \mathrm{Conv}(D)$, i.e. the orthogonal projection eliminating the auxiliary variables $v^i$ and $y^i$ yields the convex hull of the set of interest $D$. Therefore, the result follows by explicitly computing this projection, yielding a system of linear constraints equivalent to the LP relaxation of (5), i.e. (5a-5d).

Use the three equations (9c), (9e), and (9g) to eliminate the $y^i$ variables. Then we may use the remaining equations (9a-9b) to eliminate $v^1$ and $v^2$, leaving the system

$$l z_1 \leq v - y + b^{(2)} z_1 - v^3 \leq b^{(2)} z_1$$

$$b^{(2)} z_2 \leq y - b^{(2)} z_1 \leq b^{(1)} z_2$$

$$b^{(1)} z_3 \leq v^3 \leq u z_3$$

$$1 = z_1 + z_2 + z_3$$

$$\boldsymbol{z} \in [0, 1]^3.$$

We may then apply the Fourier-Motkzin elimination procedure (e.g. [15, Chapter 3.1]) to project out the last remaining auxiliary variable $v^3$, giving the result. $\qquad \square$

### A.5 Proof of Proposition 4

*Proof.* Parameterize the sequence of instances by $i$. For each $i$, define $\boldsymbol{w}^{i,1} = (\sqrt{1 - i^{-2}}, i^{-1})$, $\boldsymbol{w}^{i,2} = (-\sqrt{1 - i^{-2}}, i^{-1})$, $b_i^{(1)} = 1$, and $b_i^{(2)} = 0$. Note that $||\boldsymbol{w}^{i,1}||_2 = ||\boldsymbol{w}^{i,2}||_2 = 1$, and so all the problem data is bounded in magnitude by one. The unique optimal solution to (1) is $\boldsymbol{\beta}^{i,*} = (0, i)$, giving the result. $\qquad \square$

### A.6 Proof of Proposition 5

*Proof.* The bound on objective values follows immediately from Corollary 1 and the fact that $F(b^{(1)}, b^{(2)}, l, u) \subseteq W(b^{(1)}, b^{(2)}, l, u)$ for any choice of data. Additionally, as (1) only constrains $\boldsymbol{\beta} \in X$, feasibility follows from (3d). $\qquad \square$

### A.7 Proof of Proposition 6

*Proof.* Consider the following problem instance parameterized by a positive integer $T$. Take $n = 2T$, $m = 2$, and $X = [-1, +1] \times \{+1\}$. Furthermore, for each $i \in [\![T]\!]$, define $\boldsymbol{w}^{+,i} = (T, 1 - i)$, $b_{+,i}^{(1)} = 1$, and $b_{+,i}^{(2)} = 0$. Similarly, for each $i \in [\![T]\!]$, define $\boldsymbol{w}^{-,i} = (-T, 1 - i)$, $b_{-,i}^{(1)} = 1$, and $b_{-,i}^{(2)} = 0$. From inspection, we can observe that for any $\boldsymbol{\beta} \in X$, there is at most one $i$ with

$r(\boldsymbol{w}^i \cdot \boldsymbol{\beta}; b_i^{(1)}, b_i^{(2)}) > 0$. Therefore, we can infer that the optimal reward for (1) is 1, which can be attained by setting $\boldsymbol{\beta} = (k/T, 1)$ for any $k \in \bigcup_{k=1}^T \{-k/T, +k/T\}$.

In contrast, the LP relaxation bound can be bounded below by a constant. By projecting out the auxiliary $\boldsymbol{z}$ variables from the LP relaxation (5a-5d), we can compute that the convex hull of $\mathrm{cl}(\mathrm{gr}(r(\cdot; 0, 1); [l, u]))$ is

$$Q(l, u) :=$$
$$\left\{ (v, y) \in [l, u] \times \mathbb{R}_{\geq 0} \;\middle|\; y \leq \frac{1}{1-l}(v - l), \; y \leq \frac{1}{u-1}(u - v) \right\}.$$

Furthermore, for each $i \in [\![T]\!]$ we can computer valid bounds on $v_{+,i}$ as $l_{+,i} = \min_{\beta \in X} \boldsymbol{w}^{+,i} \cdot \boldsymbol{\beta} = -T + 1 - i$ and $u_{+,i} = \max_{\beta \in X} \boldsymbol{w}^{+,i} \cdot \boldsymbol{\beta} = T + 1 - i$. Similarly, valid bounds for each $v^{-,i}$ are $l_{-,i} = -T + 1 - i$ and $u_{-,i} = T + 1 - i$. Piecing it all together, we now fix $\boldsymbol{\beta} = (0, 1)$, which due to (3b) will fix $v_{+,i} = v_{-,i} = 1 - i$ for each $i \in [\![T]\!]$. Accordingly, the largest value we may set $y_{+,i}$ such that $(v_{+,i}, y_{+,i}) \in Q(l_{+,i}, u_{+,i})$ is $y_{+,i} = \frac{T}{T+i}$. Similarly, the maximum allowed value for each $y_{-,i}$ such $(v, y) \in W(b^{(1)}, b^{(2)}, l, u)$ is satisfied is $y_{-,i} = \frac{T}{T+i}$. The reward at this LP feasible point is then

$$\frac{1}{n} \sum_{i=1}^T (y^{+,i} + y^{-,i}) = \frac{1}{n} \sum_{i=1}^T \left( \frac{T}{T+i} + \frac{T}{T+i} \right)$$
$$= \sum_{i=1}^T \frac{1}{T+i} \geq \sum_{i=1}^T \frac{1}{2T} = 1.$$

$\square$