[Reviews · NeurIPS 2020]

Review 1

Summary and Contributions: This paper considers the problem of setting a reserve price for a second price auction. For simplicity, the reserve price is a linear function of the features. First, the authors prove a hardness result which states that, unless the exponential-time hypothesis fails, there is no poly time algorithm for the reserve price optimization problem. Next, the authors give an MIP formulation of the problem as well as an LP relaxation of the the MIP formulation. Finally, the authors present some experimental results to show the effectiveness of the MIP and the LP relaxation.

Strengths: The paper is written well and easy to understand and follow. The authors clearly explain what is difficult about the problem (namely that ETH implies that the problem is difficult in general). They also run some nice experiments to illustrate the different performances of the algorithms. To the best of my knowledge, this work is novel. It is intriguing to me that setting reserve prices with features has not really been dealt with much in the literature. This paper may be of interest to practitioners looking to design better algorithms for reserve pricing.

Weaknesses: I am not sure the paper is contributing much from an ML standpoint. They propose a MIP and its LP relaxation then runs some experiments to showcase the different quality of solutions amongst different methods. I think the present paper may be more suited for a more specialized venue for MIP or for auctions, like WWW.

Correctness: The authors assert some claims which all seem sound to me. However, I have not checked the proofs int he appendix.

Clarity: The paper is written clearly.

Relation to Prior Work: The paper clearly discusses how the present work differs from previous works.

Reproducibility: Yes

Additional Feedback: It is interesting that the LP does not really do better than constant prices for synthetic data. Is this just because of the nature of the synthetic data? LP seems to be clearly better than constant prices for ebay experiments though. Minor Comments: - I found the explanation of (2) confusing. In line 39, it says that the setting recovers first price auction and pure price-setting. Do you mean by setting artificial bids? I understood (2) as: "the revenue that one gets given a reserve price v, highest bid b^1 and second highest bid b^2". ===== Edit: I have looked at the author's response and the other reviews. I am happy to accept that there may be a good proportion of attendees that I may be interested in such work so I have increased my score.


Review 2

Summary and Contributions: In the second-price auction with reserve price, the publisher sets a reserve price before the auction, and the payment of the winner is defined as the maximum of the second highest bid and the reserve price. The paper deals with the problem of deciding the reserve price from the side information to maximize the payment from the winner. It considers using linear regression to infer the best reserve price. Then the problem is reduced to the optimization problem of optimizing the model parameters so as to maximize the payment. The main contribution of this paper is to discuss how to solve this optimization problem. The objective function of the optimization problem is complicated, and so the optimization seems hard. Indeed, the paper proves that the k-densest subgraph can be reduced to the problem, meaning that there is no polynomial-time algorithm for it under a suitable assumption. Then, the paper proposes a mixed-integer programming (MIP) formulation. MIP is an NP-hard problem, but several efficient solvers are available for it. A linear programming relaxation for the MIP formulation can be also used. These approaches are evaluated by computational experiments. The result show that the algorithms of using MIP are superior to other approaches using the LP relaxation or other algorithms proposed by previous studies.

Strengths: - It formulates a new interesting optimization problem, which naturally appears in realistic setting of auctions. - The proposed approach is efficient, both in its performance (as shown by the experiments) and in the usability (since several efficient MIP solvers are available). - It also revels the computational difficulty of the optimization problem. - The topic is closely relevant to the NeurIPS community. The paper deals with an application of the machine learning to the common auction setting. It must be also interesting to the optimization community.

Weaknesses: Although I do not agree, there might be an opinion that the contribution is not enough because the paper just formulates a hard optimization problem as MIP, which is another hard optimization problem.

Correctness: As far as I checked, the claims and method seems correct. Since proofs of several claims are not presented, I could not check all the details.

Clarity: The paper is written very well. It is easy to read. It would be better if important claims (e.g., Theorem 1 and Proposition 2) were presented.

Relation to Prior Work: The paper clearly dicuss the difference from the previous studies.

Reproducibility: Yes

Additional Feedback: L.63: is a very popular assumption is computational complexity -> is a very popular assumption in computational complexity === I read the authors' feedback and my opinion hasn't changed.


Review 3

Summary and Contributions: The paper introduces a new technique for modelling reserve price in second-price auctions. The idea is to use contextual information as input variables in a linear model, and in doing so improve the reserve price and increase the profit. The authors propose a MIP optimization approach to discover model, and conduct experiments with real-world and syntetic data.

Strengths: - using contextual information improves reserved price. - solid algorithmic approach. - computational complexity results.

Weaknesses: - Limited target audience as the method is meant to be used in second-price auctions. Still the approach may be useful in this particular application. - Comparison to [38] should have been more detailed as both use contextual information. How their model is different than yours?

Correctness: The methodology is solid. The evaluation is done over a temporal data (the actions occur over time), but the training and the test is done in a cross-validation style, so it opens up a door to a data leakage (=predicting current auction from a model trained using future auctions). This probably doesn't have that a strong effect.

Clarity: Yes.

Relation to Prior Work: A stronger discussion with [38] would have been more appropriate (though the page limit is an issue here). Are both linear models, how are the models differ?

Reproducibility: Yes

Additional Feedback: The paper can be improved further by a more thorough comparison to [38]. More careful evaluation to avoid data leakage (though this probably doesn't change the conclusions). Perhaps a more straightforward approach here is possible where one would approximate r with a smooth differentiable function.

[Author Response · NeurIPS 2020]

We thank the reviewers for their insightful comments. We are humbled that the reviewers view this work as novel (R1), that it considers an interesting problem that is closely relevant to the NeurIPS community (R2), and that our algorithmic approach and methodology are "solid" (R4). We spent considerable effort to make the writing and structure of the paper easy to read and follow, and so are grateful that this effort was reflected in the referees feedback on clarity (R1, R2, R4). To recap, we *develop algorithms* to solve a *learning problem* in *online advertising auctions*. Our optimization algorithms are capable of producing models that perform better than those produced by existing techniques.

**Contribution to ML.** R1 expressed concerns about the contribution of the work "from an ML standpoint". While we do not deal directly with, e.g., statistical properties of our learned models, our work is firmly grounded in an ML application. NeurIPS has historically made a home for such papers that broadly define ML. To pick just one example, an Honorable Mention for the Outstanding Paper Award at NeurIPS 2019 (*Fast and Accurate Least-Mean-Squares Solvers* by Maalouf, Jubran, and Feldman) is a "pure algorithms" paper on a crucial subroutine in ML training algorithms.

**Venue fit.** R1 stated that the paper "may be more suited for a more specialized venue for MIP or for auctions, like WWW". We agree that this work is likely be of interest to specialized communities in auctions and discrete optimization. However, both audiences are well-represented in the NeurIPS community, and indeed R2 states that "The topic [of this paper] is closely relevant to the NeurIPS community".

**Connection with Mohri and Medina (2016).** R4 expressed a desire for a more detailed explanation of the connection between our proposed method and the work of Mohri and Medina (2016); referred to MM to follow. R4 asks: "Are both papers considering linear models"?, and the answer is: **Yes**. Both papers consider training over an identical linear model with discontinuous loss. MM presents an algorithm for the training problem by smoothing the loss using a surrogate function which thus does not guarantee global optimality to the original problem. Our novel contribution is a family of algorithms which, as the computations show, can close the optimality gap of the MM method *to ultimately produce better performing models*.

**Second-price auctions have limited target audience.** R4 expresses concern that the paper may have a "limited target audience as the method is meant to be used in second-price auctions". We respectfully disagree: generalized second-price (Vickrey) auctions are commonly used by online ad platforms, and have been extensively studied, including at past NeurIPS. By our count, 5 papers at NeurIPS 2019 included "auction" in the title, and two of these directly consider second-price auctions or generalizations thereof. We also highlight that, as noted in the introduction, our approach can be slightly modified to handle other auction variants, such as first price auctions or pure price-setting problems.

**Why does LP do poorly on synthetic data?** R1 states that "it is interesting that the LP does not really do better than constant prices for synthetic data", and then asks why this might be the case. The explanation is: The models handle constant offsets in slightly different ways. The constant policy (CP) does not enforce bounds on the magnitude on the offset term, whereas other models do. Therefore, the CP model was not actually attainable by LP. This does not matter in high dimensions (eBay), but can make a difference on our low dimensional synthetic data. We have now implemented models that tune over the magnitude bounds for the offset, and can verify that i) LP does substantially better on the synthetic data (beating CP), and ii) the results on the eBay data set remain unchanged. We are currently repeating the synthetic data experiments on higher dimensional data to mitigate the spurious impact of the constant offset. We thank R1 for their great observation, which lead to a substantial improvement of the performance of LP.

**Data leakage.** We thank R4 for pointing out the potential for temporal data leakage. To provide more detail, we are not doing validation on "future" data: our validation set is constructed by randomly holding out a portion of the training data. Ultimately, we agree with R4 that this likely does not have a strong effect on the results.

**Why not approximate $r$ with a smooth differentiable function?** R4 suggests, as an alternative approach, to "approximate $r$ with a smooth differentiable function." This is an interesting and natural idea! We are considering this approach in ongoing work, and indeed MM studied a variant of this smoothing technique. One thing to keep in mind is that the problem remains nonconvex even after smoothing, and so is still NP-hard. Practically, the natural way to solve this problem would be using (stochastic) gradient ascent. However, we believe that the performance would not be materially different from running gradient ascent directly on the discontinuous problem, which can be observed to have very poor performance in Tables 1 and 2. In a sentence, we believe this because i) the gradients only differ substantially in a small neighborhood around discontinuities, and ii) even after smoothing the problem is very likely still have many (bad) local minimums (see Figure 1).

**Presentation of important claims.** R2 stated that "it would be better if important claims...were presented." We may be misunderstanding this comment, but we interpret it to mean that R2 is asking for proofs of the claims. In the submission, we rigorously proved each of the claims in the supplementary materials. To make this connection clear, we have added proof "stubs" to follow each formal result, pointing to the specific subsection in the appendix containing the proof.

[Meta-Review · NeurIPS 2020]

The paper proposes a new MIP formulation for setting reserve prices in in auctions. While the focus of the work is on the edges of the interest of NeurIPS, the paper is well-written and the algorithmic approach is novel and solid. The new formulation results in good experimental results. We urge the authors to incorporate clarifications on the importance of their work the the ML community into the main paper, e.g. into the abstract and the introduction. The paper would also benefit from a discussion of how the proposed algorithm can be included in practical scenarios.